# Learning Curve of Robotic Lobectomy for the Treatment of Lung Cancer: How Does It Impact on the Autonomic Nervous System of the Surgeon?

**DOI:** 10.3390/jpm13020193

**Published:** 2023-01-21

**Authors:** Antonio Mazzella, Shehab Mohamed, Patrick Maisonneuve, Giulia Sedda, Andrea Cara, Monica Casiraghi, Francesco Petrella, Stefano Maria Donghi, Giorgio Lo Iacono, Lorenzo Spaggiari

**Affiliations:** 1Division of Thoracic Surgery, IEO, European Institute of Oncology IRCCS, 20141 Milan, Italy; 2Division of Epidemiology and Biostatistics, IEO, European Institute of Oncology IRCCS, 20141 Milan, Italy; 3Department of Oncology and Hemato-Oncology, University of Milan, 20122 Milan, Italy

**Keywords:** learning curve, lobectomy, robotic surgery, robotic-assisted surgery, RATS, surgical stress, lung cancer, thoracic surgery

## Abstract

Objective: Our purpose is to define the learning curve for robot-assisted thoracoscopic surgery lobectomy by reporting the experience of a single surgeon. Material and methods: We progressively collected the data concerning the surgical performance of a single male thoracic surgeon, from the beginning of his robotic activity as first operator from January 2021 to June 2022. We evaluated several pre-, intra- and postoperative parameters concerning patients and intraoperative cardiovascular and respiratory outcomes of the surgeon, recorded during surgical interventions, in order to evaluate his cardiovascular stress. We used cumulative sum control charts (CUSUM) to analyze the learning curve. Results: A total of 72 lung lobectomies were performed by a single surgeon in this period. Analyzing the CUSUM of several parameters, the inflection point identifying the transition beyond the surgeon learning phase was reached at cases 28, 22, 27 and 33 when considering operating time, mean heart rate, max heart rate and mean respiratory rate, respectively. Conclusions: The learning curve for robotic lobectomy seems to be safe and feasible with a correct robotic training program. The analysis of a single surgeon from the beginning of his robotic activity demonstrates that confidence, competence, dexterity and security are achieved after about 20–30 procedures, without compromising efficiency and oncological radicality.

## 1. Introduction

Since the first report about robot-assisted lobectomy performed by the da Vinci surgical system in 2002 [1] and during the last two decades, we have witnessed a constant technological improvement with the subsequent explosion of robot-assisted thoracoscopic surgery (RATS) for the treatment of early-stage non-small cell lung cancer (NSCLC).

At present, robotic lobectomy is considered an achievable and safe procedure [2,3,4,5], with clinical and oncological outcomes comparable to those of video-assisted thoracoscopic surgery (VATS)/thoracotomy, both in early [6,7,8,9] and in advanced stages (II-IIIA NSCLC) with no differences in medium and long-term survival [10].

Up to now, a number of studies have systematically analyzed the learning curve of minimally invasive (VATS or RATS) lobectomy [11,12,13,14,15,16,17,18,19]; all of these papers considered different parameters concerning the quality, the effectiveness and the radicality of the surgical act. So far, no studies have focused on the surgeon’s wellness and comfort during the surgical act.

The surgeon’s feelings and his/her cardiovascular and respiratory response during the interventions are direct indicators of comfort, feeling at ease, dexterity and security during a surgical procedure. The “hemodynamic effect” of surgical stress for the surgeon has been sporadically evaluated in the literature. The impact of surgical stress could be measured by heart rate and heart rate variability (HRV), considered as a consequence of the autonomic nervous system (ANS) activity, regulating the time interval (milliseconds) between heartbeats [20,21,22]. Some papers relate blood pressure, heart rate, O_2_ saturation and CO_2_ production with the activation of the ANS during surgery [22,23,24], and low HRV and a higher heart rate (HR) are associated with more stressful events or more stressful interventions. 

In this light, in addition to the oncologic and surgical outcomes, the evaluation of the surgeon’s cardiovascular and respiratory parameters, detected during his/her surgical activity, could represent markers of comfort, feeling at ease, dexterity and mastery of the surgical technique.

Our purpose was to define the learning curve of robotic lobectomy to provide training guidelines for RATS lobectomy by reporting our experience using the CUSUM technique.

## 2. Materials and Methods

This is a prospective observational study, starting in January 2021 and ending in June 2022. No authorization from our Ethics Committee was required because the device used is already approved in real life for the evaluation of vital parameters, both in basal conditions and under stress. 

This study was conducted in accordance with the ethical principles of the declaration of Helsinki.

We progressively collected the data concerning the surgical performances of a single male thoracic surgeon from the beginning of his robotic activity (January 2021) as first operator in robotic surgery until June 2022.

In particular, we evaluated a panel of pre-, peri- and postoperative data concerning the interventions and the cardiovascular activity and stress of the surgeon during his surgical activity.

### 2.1. Patient Work-Up

For all patients, pretreatment work-up included total body contrast-enhanced CT scan (brain, thorax and abdomen) and positron emission tomography scan. Mediastinal nodes were considered negative at clinical staging if the short axis was less than 1 cm and there was no significant (standardized uptake value < 2.5) [18F] fluoro-2-deoxy-2-D-glucose uptake. Global spirometry (with DLCO/VA calculation) and a complete cardiological assessment were routinely performed. Preoperative histological diagnosis, if possible, was obtained by biopsy (fiberoptic bronchoscopy or fine-needle aspiration biopsy—FNAB). All patients were then discussed at our multidisciplinary meeting before the operation.

### 2.2. The Surgeon’s Previous Experience

The surgeon, the subject of the study, was a 35-year-old consultant, experienced in open surgery (>200 anatomical lung resections) and VATS (>200 minor thoracoscopic interventions, >20 VATS lobectomies). After having had an overview of the robotic system through online training, participated in the procedures as bedside assistance (about 150 cases) and learned the surgical setup, the surgeon developed the robotic skills in the simulator and began his robotic program.

### 2.3. Surgical Technique (RATS Lobectomy via Four-Arm Robotic Approach with Utility Incision)

Our technique and robotic approach have already been described [25]. All procedures were performed under general anesthesia with a double-lumen endotracheal tube. Patient and personnel were positioned as previously reported [12]. With the patient in lateral decubitus, a 3 cm utility incision was performed at the fourth or fifth intercostal space, anteriorly. Through this incision, an 8 mm 30° three-dimensional robotic endoscope was inserted into the chest to explore the pleura and help perform the other three 8 mm ports under direct vision: the camera port in the seventh or eighth intercostal space on the midaxillary line, and two other 8 mm ports at the seventh or eighth intercostal space in the posterior axillary line and at the auscultatory triangle, respectively. We used the Da Vinci XI system, and the robot was usually driven over the patient’s shoulder at a 15° angle and attached to the four ports. The surgical cart was docked from the left side of the patient either for right or left thoracic procedures; the camera could be moved between two different ports, allowing for a better view; we used the EndoWrist stapler (30 vascular and 30 or 45 parenchyma), which could be placed through a 12 mm port (for inferior lobectomy or segmentectomy, it was placed alternatively at the utility thoracotomy or posterior axillary line; for upper lobectomy, it was placed at the posterior axillary line), as previously reported [26]. The fissure-last approach was always used. Portal placement did not change with the type of resection or the side. 

Our postoperative protocol provides for early mobilization of the patient (in the first 12 h after surgery), local anesthesia with blocks or infiltrations, low use of opioids and removal of the drainage when there are no air leaks and fluids are <250 cc. 

### 2.4. Patient Outcomes

We evaluated several pre-, intra- and postoperative parameters concerning the patients (Table 1). In particular, we considered operative time (time from skin incision to the end of the surgical procedure), pathological stage (8th edition of the TNM classification), dimensions of lung tumor and hilar and mediastinal harvested lymph nodes (obtained by histologic exam), conversion (unplanned extension of the incision and rib spreading), postoperative complications (according to Clavien–Dindo classification), postoperative hospital stay (from the day of the surgical procedure to the day of discharge) and chest tube drainage duration. 

However, we must consider that many patients come from different Italian areas (sometimes very distant from this oncologic reference center), and therefore, because of logistical problems, they stay in the hospital longer, even if they dischargeable.

### 2.5. Cardiovascular and Respiratory Activity of the Surgeon 

The surgeon underwent a full cardiological check-up before starting the study. No cardiac/respiratory or general comorbidities were detected. The surgeon does not take any pharmacological therapy.

The surgeon’s cardiac and respiratory parameters were taken with a wearable device: Healer R2 (MDD Class IIa Medical Device developed by L.I.F.E. Italia S.r.l., Milano, Italy) (www.x10x.com; info@x10x.com) intended for multiparametric monitoring applications (clinical or wellbeing) and diagnostic exams (polysomnography and ECG Holter) in hospital/outpatient and home settings. 

On a sensorized garment (Healer R2), the device included a data logger for data acquisition, storage and transmission with Wi-Fi or 4G connectivity, a cloud platform for data storage and elaboration, desktop software (Healer Desktop), a web portal (Healer Cloud) and two apps (Healer R and My Healer) (Figure 1). 

The sensorized garment Healer R2 can record several physiological parameters and signals (Figure 2), each one with its own sampling frequency: -**Cardiovascular activity (mean, maximum and minimum heart rate)**, thanks to 6-lead ECG signal at 500 Hz by using 4 ink-based dry electrodes;-**Respiratory activity (mean, maximum and minimum respiratory rate)**, desaturation and time of desaturation, thanks to 3-channel respiratory signal at 50 Hz from strain circumferential sensors placed at the thoracic, xiphoid and abdominal levels;-Body activity and temperature (mean, maximum and minimum body temperature) thanks to a contact sensor under the right armpit;-Blood oxygen saturation (SpO2) (mean, maximum and minimum SpO2, desaturation and time in desaturation) from an optical module under the left armpit;-**Activity level and body position** from an inertial measurement unit (IMU) on the back.

We did not consider blood pressure as a parameter for various reasons: first of all, mounting and pumping by a Holter device every 10 min could in itself cause stress to the surgeon; the second aspect is the possible damage linked to the pumping to the artery; lastly, continuous and intermittent pumping could compromise the surgical act, especially if prolonged and complex.

### 2.6. Statistical Methods

We used cumulative sum control charts (CUSUM) to analyze the learning curve based on operating time, heart rate and respiratory rate. The CUSUM is the running total of differences between the individual data points and the mean of all data points with the learning curve considered ‘complete’ at the graphical peak. We used the inflection point of the CUSUM chart to identify the end of the surgeon-learning phase. We used the slope of linear regression models to quantify the change in selected parameters during the learning phase and during the ensuing phase, and the F-test to assess whether changes were significant during the two respective periods. We also compared patients’ characteristics and surgeon’s measurements of stress (heart rate, respiratory frequency, body temperature, saturation) during the learning phase and the ensuing phase, using the Fishers’ exact test for categorical variables and the Student t-test or the nonparametric median test for continuous variables. Analyses were performed with the SAS software version 9.4 (Cary, NC). All *p*-values were two-sided.

## 3. Results

A total of 72 lobectomies were performed from the beginning of robotic activity as first operator (January 2021–June 2022) by a single surgeon (AM) at the IEO thoracic surgery division.

Clinical, demographic and preoperative outcomes are reported in Table n. 1. In this period, 24 right upper lobectomies (33%), 4 middle lobectomies (5.5%), 20 right lower lobectomies (28%), 14 left upper lobectomies (19.5%) and 10 left lower lobectomies (14%) were performed.

Pathological findings revealed 50 adenocarcinomas (69.4%), 8 squamous cell carcinomas (11.1%), 1 NSCLC (1.4%), 1 adenosquamous carcinoma (1.4%), 2 large cell carcinomas (2.8%), 1 SCLC (1.4%), 7 typical carcinoids (9.7%) and 2 benign diseases (2.8%). Mainly stage IA (25 patients) and IB (35 patients) patients were operated on. In 10 cases, the stage was II or III because of occult lymph node involvement.

Only two conversions in thoracotomy were observed, because of intraoperative bleeding in one case and technical difficulties in another (inadequate pulmonary exclusion).

No 30- or 90-day mortality was observed. In 16 patients, we detected postoperative complications: 6 persistent air leaks, 7 atrial fibrillations, 1 Takotsubo syndrome, 1 recurrent nerve paralysis and 1 lung atelectasis requiring bronchoscopy aspiration (Table 1).

The surgeon’s cardiovascular and respiratory outcomes are described in Table 2. 

Analyzing the cumulative sum control charts (CUSUM) of various parameters, the inflection point identifying the transition beyond the surgeon learning phase was reached at cases 28, 22, 27 and 33 when considering operating time, mean heart rate, max heart rate and mean respiratory rate, respectively (Figure 3).

Concerning harvested lymph nodes, length of hospitalization and chest tube drainage and postoperative complications, no differences were found during the learning curve or after it (Figure 4).

## 4. Discussion

Minimally invasive approaches in lung surgery were first reported almost 30 years ago and they are nowadays widely employed by a large number of thoracic surgeons [18]. Indeed, especially in early lung cancer stages, the open approach is now considered anachronistic and associated with greater surgical inflammation, postoperative pain and postoperative dysfunction.

More specifically, the advantages of robotic surgery have been widely investigated in the recent past; nowadays, robotic lobectomy represents the gold standard in the treatment of early lung cancer, being feasible and safe, with clinical outcomes similar, or in some cases superior, to those of video-assisted thoracoscopic or open surgery [2,3,4,5,6,7,8,9]. 

In addition to the known advantages related to the postoperative course of the patient, the robotic approach also has important advantages for the comfort of the surgeon during the intervention. First of all, robotic instruments have various degrees of freedom, making it possible to replicate movements of the traditional open technique; secondly, they allow for considerable limitation of the natural tremor of a surgeon’s hands by converting movements into micromovements. Thirdly, the surgeon sits comfortably at the console, controlling the instruments while viewing the operating field in high-definition 3D. Last but not least, the console is ergonomic and adjustable and the surgeon can modify the height of the optical viewer or of the arms or pedal board. On the other hand, the absence of the tactile feedback of the robotic instruments and the effective distance of the surgeon from the operative field could represent not insignificant difficulties to overcome, especially at the beginning and during the learning curve.

The learning curve of robotic lobectomy has already been the subject of evaluation, since 2002 [12] when Veronesi et al. reported that the learning phase for the technique included the first 18 cases. Following the most recent literature, we realize that the acquisition of competence and effectiveness (learning curve) in robotic lobectomy occurs after at least 14 cases [19] and ranges between 14 and 60 cases in the various reports [12,13,14,15,16,17,18,19]. 

The most important outcomes considered in the evaluation of a learning curve are represented by variables specifically linked to the surgical intervention, such as actual operative time, intraoperative conversion rate to thoracotomy or intraoperative bleeding; other parameters concern oncological outcomes (number of harvested lymph nodes or T stages—dimensions or locally advanced tumors) or postoperative complications [12,13,14,15,16,17,18,19].

None of these studies take into direct consideration the surgeon’s comfort during surgical procedures, how the surgeon feels or reacts during the interventions or how the surgeon varies or improves his physiological and autonomic reactions in the management of surgical stress, operation after operation. Thus, we also considered hemodynamic, metabolic and respiratory responses to stress, regulated by the autonomic nervous system. In particular, stress activates the ANS sympathetic component with chronotropic and inotropic effects on the heart, vasocontraction and increasing of blood pressure, as well as of the respiratory rate, characterized by shallow breaths [24]. In this light, heart rate, heart rate variability, blood pressure, O2 saturation, and time of desaturation have been considered in various papers as markers of stress linked to surgical acts and regulated by noradrenergic pathways of sympathetic innervation on the cardiovascular and respiratory systems. [20,21,22,23,24]. In accordance with our cardiologic division, we used a wearable device (Healer R2) for monitoring the ANS sympathetic component. This device is already utilized for remote multiparametric monitoring of patients and for diagnostic exams in hospital or at home. The use of this kind of device exploded during the COVID era thanks to the possibility of controlling parameters remotely via the Cloud. We deliberately chose not to use blood pressure as a stress parameter because mounting and pumping by a Holter device every 10 min could in itself cause stress to the surgeon and compromise the surgical act, especially if prolonged and complex.

Our analysis of cumulative sum control charts (CUSUM) shows that the inflection point identifying the transition beyond the surgeon learning phase was reached after 28 (operating time), 22 (mean heart rate), 27 (max heart rate) and 33 (mean respiratory rate) cases, respectively.

No differences were found between during the learning curve and after it concerning conversions in thoracotomy, harvested lymph nodes, length of hospitalization and chest tube drainage and postoperative complications.

Mean and max heart rate are direct signs of ANS sympathetic system activation, linked to vasocontraction and chronotropic and inotropic effects on the heart. At the beginning, the surgeon does not feel comfortable with the new technique, and even normal gestures such as moving or holding back the lung parenchyma to expose the operating field become stressful, being transformed into an increased mean and maximum heart rate and mean respiratory rate. After this initial phase (22 and 27 cases for the mean and maximum heart rate and 33 cases for the mean respiratory rate), these parameters decrease in value and stabilize. This decreasing is a direct consequence of the reduced stress of the surgeon and an indirect sign that he/she has acquired competence, confidence, dexterity and security in the surgical gesture. According to the “autonomic response”, operative time decreases operation after operation, and the learning phase is reached after 28 cases.

On the basis of our analysis, we can assert that an individual surgeon gains competence and confidence after about 22–30 cases. On the other hand, the lack of significant differences concerning other outcomes (i.e., conversions to thoracotomy, harvested lymph nodes or length of hospitalization, chest tube drainage duration and postoperative complications) represents the efficiency and the oncological radicality, which are always maintained constant right from the start thanks to the presence of an experienced proctor assistant at the beginning. Indeed, there are several factors that affect the duration of the learning curve and they should be considered when implementing a robotic training program.

First of all, before starting a training program as first operator, the surgeon should already have experience with the robotic approach. In other words, the trainee should initially acquire the basic robotic skills by online training or using the simulator [27].

The trainee must also participate in the interventions, as a bedside assistant to an experienced proctor, in order to learn and implement the skills previously acquired on the simulator. During this phase, it is possible to use the dual console, allowing the console surgeon to perform the surgery simultaneously with a proctor, in order to complete small parts of the intervention, such as the isolation of a pulmonary vein or artery.

Needless to say, the training program should take place in a high-volume center, with the mentoring of an experienced proctor, especially for the first cases.

The OR team around the surgeon represents another fundamental issue. Nurses, assistants and anesthesiologists must have the right skills to manage the novel technique. 

Another aspect to consider for the development and the learning of a technique is the personality of the training and of the teaching surgeon. Sier et al. [28] concluded, in general, that surgically-interested individuals demonstrated higher levels of conscientiousness, openness and extraversion and lower levels of neuroticism. I would add that during a learning curve, the humility of accepting advice or seeking help in complex situations that are dangerous for the patient is mandatory. On the other hand, the teaching surgeon should present an available and pedagogical personality during this phase. 

In light of these latter statements, the personality and characteristics of each individual surgeon may vary. This aspect could significantly affect the learning curve, improving it, or in some cases, making it much longer.

One “case” is not enough to make a final conclusion, as there could be many surgical personalities, and therefore results could be different, and the mental characteristics of each surgeon could substantially influence the results related to the responses of the autonomic nervous system. The next step of this line of research could include a collaboration with psychologists and psychiatrists in order to explore the effects of surgery on various surgeons, each characterized by different personalities.

In conclusion, the learning curve for robotic lobectomy seems to be safe and feasible with a correct robotic training program, taking into account the different surgeons’ characteristics. The analysis of a single surgeon, from the beginning of his robotic activity, shows that confidence, competence, dexterity and security are achieved after 20–30 procedures, without compromising efficiency and oncological radicality.

## Figures and Tables

**Figure 1 jpm-13-00193-f001:**
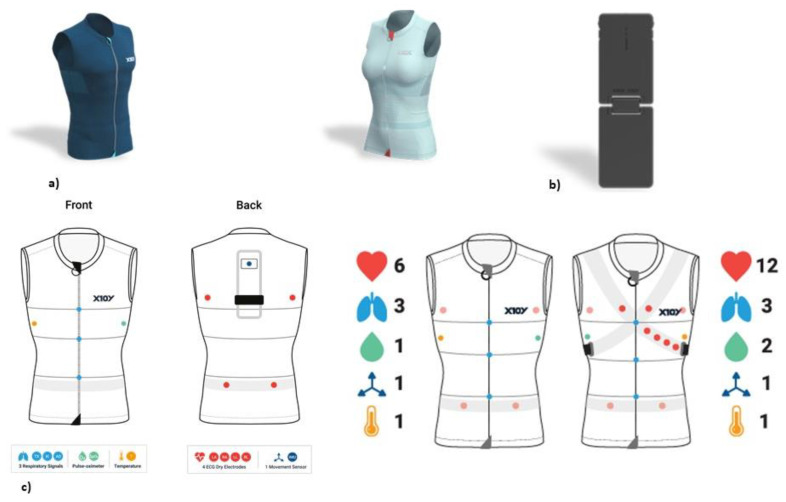
(**a**) Healer R2 wearable devices (male and female models); (**b**) logger KoR 1, connecting to the back of the garment for storing the data and transferring it via 4G/5G/Wi-Fi; (**c**) embedded sensors to detect heart rate, respiratory rate, temperature, saturation and desaturation.

**Figure 2 jpm-13-00193-f002:**
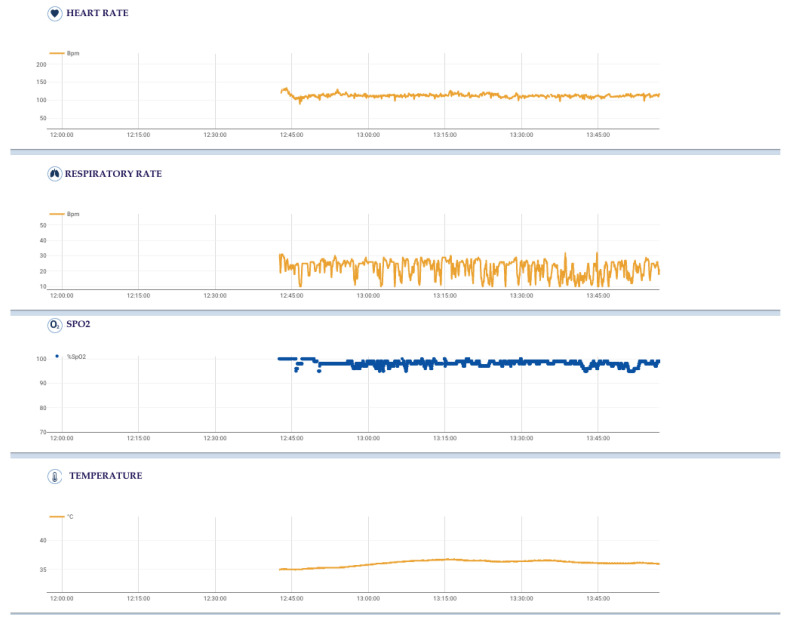
Physiological parameters, each with its own sampling frequency, acquired by sensorized garment Healer R2 and elaborated by Healer software of a single intervention (start time h: 12:38, end time h 14:00).

**Figure 3 jpm-13-00193-f003:**
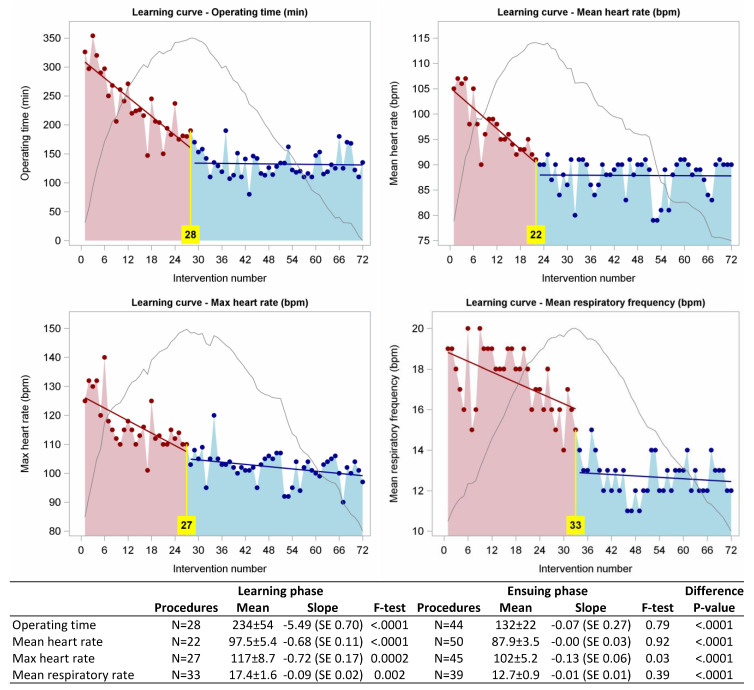
Learning curves for robotic-assisted lobectomy concerning operative time, mean heart rate, max heart rate and mean respiratory frequency. The grey lines represent the cumulative sum control charts, and the value in the yellow box the number of procedures at the graphical peak, representing the end of the learning phase. The regression lines highlight trends in the training phase (red) and in the ensuing phase (blue).

**Figure 4 jpm-13-00193-f004:**
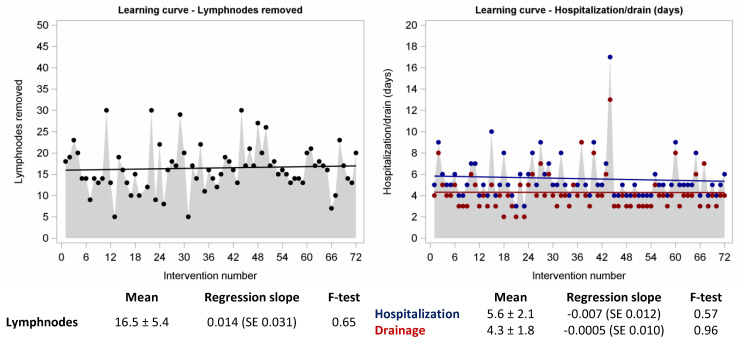
Number of harvested lymph nodes and length of hospitalization/drainage in patients who underwent robotic-assisted lobectomy.

**Table 1 jpm-13-00193-t001:** Demographic, clinical and pathological outcomes of the patients.

Characteristics	All Procedures	First 30 Procedures	Next 42 Procedures	*p*-Value
	N (%)	N (%)	N (%)	
	72 (100.)	30 (100.)	42 (100.)	
Sex				
Male F	30 (41.7)	13 (43.3)	17 (40.5)	
Female M	42 (58.3)	17 (56.7)	25 (59.5)	0.81
Age				
<60	18 (25.0)	11 (36.7)	7 (16.7)	
60–69	33 (45.8)	11 (36.7)	22 (52.4)	
≥70	21 (29.2)	8 (26.7)	13 (31.0)	0.28
Comorbidities *				
No	15 (20.8)	2 (6.7)	13 (31.0)	
Yes	57 (79.2)	28 (93.3)	29 (69.0)	0.02
Lung	23 (31.9)	9 (30.0)	14 (33.3)	0.80
Cardiac	38 (52.8)	14 (46.7)	24 (57.1)	0.47
Metabolic	35 (48.6)	13 (43.3)	22 (52.4)	0.48
Other	26 (36.1)	10 (33.3)	16 (38.1)	0.80
Side				
Right	48 (66.7)	23 (76.7)	25 (59.5)	
Left	24 (33.3)	7 (23.3)	17 (40.5)	0.20
Lobe				
Inferior	30 (41.7)	12 (40.0)	18 (42.9)	
Medial	4 (5.6)	4 (13.3)	0 ( 0.0)	
Superior	38 (52.8)	14 (46.7)	24 (57.1)	0.06
Stage				
IA	25 (34.7)	11 (36.7)	14 (33.3)	
IB	35 (48.6)	9 (30.0)	26 (61.9)	
II-III	10 (13.9)	8 (26.7)	2 (4.8)	
Benign	2 (2.8)	2 (6.7)	0 (0.0)	0.004
Grade				
G1	13 (20.0)	3 (10.0)	10 (23.8)	
G2	33 (50.8)	14 (46.7)	19 (45.2)	
G3	19 (29.3)	8 (26.7)	11 (26.2)	0.52
Histology				
ADK	50 (69.4)	19 (63.3)	31 (73.8)	
SCC	8 (11.1)	3 (10.0)	5 (11.9)	
Adenosquamous	1 (1.4)	1 (3.3)	0 (0.0)	
NSCLC	1 (1.4)	1 (3.3)	0 (0.0)	
Large cell	2 (2.8)	0 (0.0)	2 (4.8)	
SCLC	1 (1.4)	0 (0.0)	1 (2.4)	
Carcinoid	7 (9.7)	4 (13.3)	3 (7.1)	
Benign	2 (2.8)	2 (6.6)	0 (0.0)	0.36
Dimension (mm)				
Mean ± SD	22.7 ± 9.0	24.9 ± 9.8	21.1 ± 8.1	0.07
Median (range)	21 (7–49)	23.5 (7–49)	21 (9–38)	0.12
Harvested lymphnodes				
Mean ± SD	16.5 ± 5.4	16.2 ± 6.3	16.6 ± 4.8	0.76
Median (range)	16 (5–30)	15 (5–30)	16.5 (5–30)	0.42
Conversion				
No	70 (97.2)	29 (96.7)	41 (97.6)	
Yes	2 (2.8)	1 (3.3)	1 (2.4)	1.00
Complication *				
No	55 (77.5)	21 (70.0)	34 (81.0)	
Yes	16 (22.5)	9 (30.0)	7 (16.7)	0.25
Air leak	6 (8.3)	2 (6.7)	4 (9.5)	1.00
AF/arrhythmia	8 (11.1)	5 (16.7)	3 (7.1)	0.26
Other	3 (4.2)	3 (10.0)	0 (0.0)	0.07
Hospital stay (days)				
Mean ± SD	5.6 ± 2.1	5.7 ± 1.8	5.5 ± 2.3	0.77
Median (range)	5 (3–17)	5 (3–10)	5 (4–17)	0.16
Drain (days)				
Mean ± SD	4.3 ± 1.8	4.2 ± 1.4	4.4 ± 2.0	0.70
Median (range)	4 (2–13)	4 (2–8)	4 (3–13)	0.37

* a single patient may report more than 1 comorbidity or develop more than 1 complication.

**Table 2 jpm-13-00193-t002:** Surgeon’s cardiovascular and respiratory outcomes.

Characteristics	All Procedures	First 30 Procedures	Next 42 Procedures	*p*-Value
	N (%)	N (%)	N (%)	
	72 (100.)	30 (100.)	42 (100.)	
Mean heart rate				
Mean ± SD	90.8 ± 6.1	95.0 ± 6.3	87.8 ± 3.6	<0.0001
Median (range)	90 (79–107)	94.5 (84–107)	89 (79–91)	<0.0001
Min heart rate				
Mean ± SD	82.3 ± 7.7	87.7 ± 6.9	78.5 ± 5.6	<0.0001
Median (range)	82 (68–98)	89 (75–98)	80.5 (68–85)	0.005
Max heart rate				
Mean ± SD	107.6 ± 9.8	115.6 ± 9.0	101.8 ± 5.3	<0.0001
Median (range)	105 (90–140)	113.5 (101–140)	102 (90–120)	<0.0001
Mean respiratory rate				
Mean ± SD	14.8 ± 2.7	17.6 ± 1.6	12.9 ± 1.3	<0.0001
Median (range)	14 (11–20)	18 (14–20)	13 (11–17)	<0.0001
Min respiratory rate *				
Mean ± SD	10.2 ± 1.1	10.3 ± 1.5	10.1 ± 0.7	0.55
Median (range)	10 (8–13)	10 (8–13)	10 (8–12)	0.92
Max respiratory rate				
Mean ± SD	23.9 ± 2.0	24.2 ± 1.6	23.7 ± 2.3	0.27
Median (range)	24 (21–33)	24 (21–28)	23.5 (21–33)	0.02
Mean body temperature				
Mean ± SD	36.4 ± 0.2	36.4 ± 0.2	36.4 ± 0.3	0.49
Median [range]	36.3 [35.6–36.9]	36.5 [36.1–36.7]	36.3 [35.6–36.9]	0.15
Min body temperature				
Mean ± SD	36.0 ± 0.5	36.1 ± 0.3	35.9 ± 0.5	0.11
Median (range)	36.0 (34.5–37.1)	36.1 (35.2–37.1)	35.9 (34.5–36.8)	0.11
Max body temperature				
Mean ± SD	36.8 ± 0.3	36.8 ± 0.3	36.8 ± 0.3	0.82
Median (range)	36.8 (35.7–37.4)	36.8 (35.7–37.2)	36.8 (36.3–37.4)	0.24
Mean saturation				
Mean ± SD	98.1 ± 0.2	98.1 ± 0.3	98.0 ± 0.2	0.22
Median (range)	98 (98–99)	98 (98–99)	98 (98–99)	0.17
Min saturation				
Mean ± SD	91.4 ± 2.6	91.1 ± 2.3	91.7 ± 2.8	0.30
Median (range)	92 (84–96)	91 (84–96)	92 (84–95)	0.03
Max saturation				
Mean ± SD	98.7 ± 0.5	98.8 ± 0.4	98.6 ± 0.5	0.05
Median (range)	99 (98–99)	99 (98–99)	99 (98–99)	0.05
Desaturation *				
Mean ± SD	69.9 ± 41.2	57.1 ± 23.4	79.1 ± 48.4	0.001
Median (range)	60 (17–243)	54 (20–146)	71 (17–243)	0.01
Mean desaturation				
Mean ± SD	3.0 ± 0.7	2.7 ± 0.6	3.3 ± 0.7	0.0002
Median (range)	3 (2–5)	3 (2–4)	3 (2–5)	0.0003

***** Desaturation is inversely correlated with mean respiratory frequency (Pearson r = −0.23 *p* = 0.048).

## Data Availability

The original contributions presented in the study are included in the article.

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
