# Peer review of "Learning Curve of Robotic Lobectomy for the Treatment of Lung Cancer: How Does It Impact on the Autonomic Nervous System of the Surgeon?"

_jpm, 2023, doi:10.3390/jpm13020193_

Round 1

Reviewer 1 Report

This article is interesting but it has some weak points that need to be resolved to make a better paper. 

Previous surgical experience is also important in a junior.  Is the surgeon a resident or a junior consultant? How many open procedures have been performed prior starting this study? 

The presence of a senior surgeon in theatre could calm one young surgeon but another with a  different personality could develop more stress. I think that surgical personality is important (1), and should be evaluated before the study. 

The authors should include in the discussion that one "case" is not enough to make final conclusion as there could be many surgical personalities,  and therefore results could be different. Could male and female have a different attitude and "surgical" personality? or its is the same? Possible sex differences should also be discussed

For example it seems evident that Heart rate and breathing rate could be associated with surgical personality or sex difference. 

I think that the paper should be expanded with another study which takes in consideration different personality traits. 

Why the length of hospital stay is so long? It is almost 6 days. Do you use ERAS protocol? 

1) Vincent Q. Sier, et al  Exploring the surgical personality, The Surgeon, 2022, https://doi.org/10.1016/j.surge.2022.01.008.

Thank you for sending this paper to JPM

Author Response

REVIEWER N. 1

This article is interesting but it has some weak points that need to be resolved to make a better paper. 

Question n. 1

Previous surgical experience is also important in a junior. Is the surgeon a resident or a junior consultant? How many open procedures have been performed prior starting this study? 

I really thank the reviewer for this remark. The surgeon was a 35-years consultant; in the text (Paragraph 2.2 The surgeon’s previous experience) is described the experience of the surgeon, priori starting study:

“The surgeon, the subject of the study, was experienced in open surgery (> 200 anatomical lung resections) and VATS (> 200 minor thoracoscopic interventions, >20 VATS lobectomies). After having had an overview of the robotic system through online training, participated in the procedures as bedside assistance (about 150 cases), and learned the surgical setup, the surgeon developed the robotic skills in the simulator and began his robotic program”

We add into the text the qualification of the surgeon and specified that he is a young consultant

 Question n. 2

The presence of a senior surgeon in theatre could calm one young surgeon but another with a different personality could develop more stress. I think that surgical personality is important (1), and should be evaluated before the study. 

I truly thank the reviewer for this remark. I really agree with him on the question of the individual surgeon's personality. As explained by the reference, indicated by the reviewer, “it became apparent that, in general, surgically-interested individuals (i.e., medical students, residents, and surgeons) demonstrated higher levels of conscientiousness, openness and extraversion, and lower levels of neuroticism, as compared to their non-surgically-interested peers”.  We add this aspect into discussion.

Another aspect to consider for the development and the learning of a technique, is the personality of the training and of the teaching surgeon. Sier et al (28) concluded, in general, surgically-interested individuals demonstrated higher levels of conscientiousness, openness and extraversion, and lower levels of neuroticism. I would add that, during a learning curve, is mandatory the humility of accepting advice or seeking help in complex situations, dangerous for the patient. On the other hand, the teaching surgeon should present an available and pedagogical personality during this phase

Question n. 3

The authors should include in the discussion that one "case" is not enough to make final conclusion as there could be many surgical personalities, and therefore results could be different.

Thanks for this observation. We add into discussion this concept

In light of these latter statements, the personality and characteristics of each individual surgeon may vary. This aspect could significantly affect the learning curve, improving it, or in some cases, making it much longer. One "case" is not enough to make final conclusion as there could be many surgical personalities, and therefore results could be different”

Question n. 4

Could male and female have a different attitude and "surgical" personality? or its is the same? Possible sex differences should also be discussed. For example, it seems evident that Heart rate and breathing rate could be associated with surgical personality or sex difference. I think that the paper should be expanded with another study which takes in consideration different personality traits. 

Thank you for this suggestion; after this remark, in collaboration with psychologists and psychiatrists we could explore the effects of surgery on various personality traits.

In my opinion, sex alone, cannot influence the learning curve. Men and women can certainly have a different basic frequency, but in this case, we are talking about a reduction in frequency (heart rate, respiratory etc etc) over time, regardless of the starting frequency. Instead, I totally agree with the reviewer, because physical and mental characteristics of each surgeon could substantially influence the results related to the responses of the autonomic nervous system.

We add this aspect into discussion

“The next step of this line of research, could include a collaboration with psychologists and psychiatrists, in order to explore the effects of surgery on various surgeons, each characterized by different personalities”

Question n. 5

Why the length of hospital stay is so long? It is almost 6 days. Do you use ERAS protocol? 

The length of hospital stay is 5 days (median) and 5.6 (mean). We must consider that the surgeon works in an Italian reference center. Many patients come from different Italian areas (sometimes very distant from Milan) and therefore, because of logistical problems, they stay in the hospital longer, even if they dischargeable.

We add this aspect into material and methods section

Our postoperative protocol provides for early mobilization of the patient (in the first 12 hours after surgery), local anaesthesia with blocks or infiltrations, low use of opioids and removal of the drainageIn a standard lobectomy we remove the drainage when there are no air leaks and fluids are < 250 cc.

We add this aspect into material and methods section

Reviewer 2 Report

The methodology of this study is very unique. However, these parameters of achievement of the robotic surgery are not so useful as general parameters,  such as an operative time or the incidence of complications, and so on. In addition, It is not appropriate to make a conclusion by one object experiment.

Author Response

REVIEWER n. 2

The methodology of this study is very unique. However, these parameters of achievement of the robotic surgery are not so useful as general parameters, such as an operative time or the incidence of complications, and so on. In addition, It is not appropriate to make a conclusion by one object experiment.

I really appreciate this remark. As the reviewer suggested, every surgeon has a different personality or stress management. In some cases, it’s very difficult to objectivate these parameters. Other more objective parameters are operative time, incidence of complications or conversion rate. These aspects have been considered in this work, as you can see reading the paper. However, I agree with the reviewer, it’s not appropriate to make a conclusion by one object experiment.

We added an important paragraph into discussion about this aspect.

“Another aspect to consider for the development and the learning of a technique, is the personality of the training and of the teaching surgeon. Sier et al(28) concluded, in general, surgically-interested individuals demonstrated higher levels of conscientiousness, openness and extraversion, and lower levels of neuroticism. I would add that, during a learning curve, is mandatory the humility of accepting advice or seeking help in complex situations, dangerous for the patient. On the other hand, the teaching surgeon should present an available and pedagogical personality during this phase.

In light of these latter statements, the personality and characteristics of each individual surgeon may vary. This aspect could significantly affect the learning curve, improving it, or in some cases, making it much longer.

One "case" is not enough to make final conclusion as there could be many surgical personalities, and therefore results could be different. and mental characteristics of each surgeon could substantially influence the results related to the responses of the autonomic nervous system. The next step of this line of research, could include a collaboration with psychologists and psychiatrists, in order to explore the effects of surgery on various surgeons, each characterized by different personalities.

In conclusion, the learning curve for robotic lobectomy seems to be safe and feasible with a correct robotic training program, taking into account the different surgeon’s characteristic.”

Round 2

Reviewer 1 Report

Thank you for answering the questions.

Thank you for sending this paper to JPM

Reviewer 2 Report

The attempt to measure the learning level of the robotic surgery by reduction of the surgeon's stress is unique. It is necessary to try it in more objects to examine the effectiveness by this method.